

# Bone marrow mesenchymal stem cell-derived exosomal microRNA regulates microglial polarization

Xianwei Huang[*], Xiong Liu[*], Jiaqi Zeng, Penghui Du, Xiaodong Huang and Jiyan Lin

Department of Emergency, Xiamen Key Laboratory for Clinical Efficacy and Evidence-Based Research of Traditional Chinese Medicine, The First Affiliated Hospital of Xiamen University, Xiamen, China

[*] These authors contributed equally to this work.

## ABSTRACT

**Objective**. This study aimed to explore the effects of bone marrow mesenchymal stem cell (BMSC)-derived exosomal miR-146a-5p on microglial polarization and the potential underlying mechanisms in oxygen-glucose deprivation (OGD)-exposed microglial cells.

**Methods**. Exosomes were isolated from BMSCs, and their characteristics were examined. The effects of BMSC-derived exosomes on microglial polarization were investigated in OGD-exposed BV-2 cells. Differentially expressed miRNAs were identified and their biological function was explored using enrichment analyses. The regulatory role of miR-146a-5p in microglial polarization was studied via flow cytometry. Finally, the downstream target gene Traf6 was validated, and the role of the miR-146a-5p/Traf6 axis in modulating microglial polarization was investigated in OGD-exposed BV-2 cells.

**Results**. BMSC-derived exosomes were successfully isolated and characterized. A total of 10 upregulated and 33 downregulated miRNAs were identified. Exosomal treatment resulted in significant changes in microglial polarization markers. miR-146a-5p was found to be significantly downregulated in OGD-exposed microglial cells treated with exosomes. Manipulation of miR-146a-5p expression modulated microglial polarization. Moreover, the miR-146a-5p/Traf6 axis regulated microglial polarization.

**Conclusion**. Our findings demonstrate that BMSC-derived exosomal via miR-146a-5p modulates microglial polarization by targeting Traf6, providing a potential thermal target for the treatment of neurological diseases involving microglial activation.

## INTRODUCTION

Microglia are the innate immune cells of the central nervous system (CNS) that surveil the microenvironment and maintain CNS homeostasis (*Prinz, Jung & Priller, 2019*). In normal conditions, microglia exhibit a ramified morphology, constantly surveying their surroundings and rapidly responding to any changes or injuries in the brain (*Borst, Dumas & Prinz, 2021*). However, in response to pathological stimuli such as ischemia, trauma, or neurodegeneration, microglia can rapidly transform into an activated and reactive

Corresponding author
Jiyan Lin, happylinjiy@163.com

phenotype with an amoeboid morphology and release pro-inflammatory cytokines, reactive oxygen species, and other neurotoxic substances that can aggravate CNS pathology (*Colonna & Butovsky, 2017*; *Kwon & Koh, 2020*). Therefore, modulating the polarization of microglia from a pro-inflammatory state to an anti-inflammatory state is an attractive therapeutic strategy for neurodegenerative diseases.

Exosomes are small extracellular vesicles (30–150 nm) that are secreted by a variety of cell types and have gained increasing attention for their functions in intercellular communication and regulation of various physiological and pathological processes (*Ran et al., 2022*). Studies have shown that exosomes derived from bone marrow mesenchymal stem cells (BMSCs) possess significant therapeutic potential in various neurological disorders, including stroke (*Li et al., 2023b*; *Xiao et al., 2023*). BMSC-derived exosomes have been shown to modulate inflammation, promote angiogenesis, and enhance neurogenesis (*Li et al., 2019*). However, the underlying molecular mechanisms by which BMSC-derived exosomes modulate microglial conversion remain elusive.

Extracellular vesicles, including exosomes, have emerged as important mediators of intercellular communication as they transfer various bioactive molecules, such as proteins, lipids, and nucleic acids between cells. Among these molecules, miRNAs are small non-coding RNAs that play a significant role in regulating gene expression and cellular processes (*Wu et al., 2022*). Exosomal miRNAs have been implicated in several biological processes, including immune response, cell differentiation, and cancer progression (*Bi et al., 2022*). Furthermore, recent studies have demonstrated the importance of exosomal miRNAs in regulating microglial polarization and neuroinflammation. For example, exosomal miR-124 has been shown to inhibit microglial activation and promote the M2-like phenotype in the central nervous system (CNS) (*Liu, Li & Song, 2022*). Additionally, exosomal miR-146a has been reported to attenuate neuroinflammation and oxidative stress in the CNS by modulating microglial polarization (*Kubota et al., 2018*; *Sun et al., 2023*). These findings suggest that exosomal miRNAs could be potential therapeutic targets for neuroinflammatory disorders treatment. However, there is still much to learn about how exosomal miRNAs regulate microglial polarization, and requires further investigation.

Here, we isolated exosomes from BMSCs and characterized their miRNA content. We then investigated the effects of exosomes on microglial polarization and activation and identified miRNAs that are involved in these processes. Our findings provided new insights into the therapeutic potential of exosomes for neurodegenerative diseases.

## MATERIALS & METHODS

### Cell culture

BV-2 cells and BMSCs from C57BL/6 mice were purchased from Procell (#CL-0493A; Procell, Wuhan, China) and Cyagen Biosciences, Inc. (#MUBMX-01001; Cyagen Biosciences, Inc., Guangzhou, China), respectively. The cells were cultivated in DMEM (Gibco, Billings, MT, USA) supplemented with 10% fetal bovine serum and 1% penicillin-streptomycin at 37 °C with 5% $CO_2$. Non-adherent cells were discarded after 3 days, and the medium was changed every 2–3 days. The cells were sub-cultured after detachment with trypsin-EDTA (Gibco) upon reaching 80–90% confluence.

### *Exosome isolation*

Exosomes were extracted from the BMSCs using the ExoQuick-TC Exosome Precipitation Solution (System Biosciences, Palo Alto, CA, USA). To remove cells and debris, BMSCs were collected and centrifuged at 2,000 g for 10 min. The ExoQuick-TC solution was then combined with the supernatant in a fresh tube. The mixture was incubated overnight at 4 °C, followed by centrifugation at 1,500 g for 30 min. The exosomes were purified by centrifuging the resultant pellet at 100,000 g for 1 h while resuspending it in phosphate-buffered saline (PBS).

## Characterization of exosomes

The morphology of the exosomes was examined using transmission electron microscopy (TEM). A drop of exosome suspension was dropped on a copper grid that had been coated with carbon and adsorbed for 5 min. Next, distilled water was utilized to clean the grid before it was negatively stained for 1 min with 2% uranyl acetate. Images were captured using a TEM (JEM-1400 Plus; JEOL Ltd, Tokyo, Japan).

Particle size analysis was performed using a Zetasizer Nano ZS (Malvern Panalytical Ltd, Worcestershire, UK). Exosome suspension was diluted with PBS and measured at room temperature.

The search for exosome markers was carried out using Western blot analysis. Briefly, exosome suspension was lysed with RIPA buffer (Beyotime, Shanghai, China). A BCA Protein Assay Kit (Beyotime) was employed to calculate the protein concentration. SDS-PAGE was performed for separating equal amounts of protein, which was then deposited onto polyvinylidene fluoride (PVDF) membranes (Millipore, Billerica, MA, USA). After being blocked with 5% skim milk for 1 h at room temperature, The membranes were then incubated with primary antibodies TSG101 (1:1,000; ab125011, Abcam, Cambridge, UK), CD9 (1:1,000; ab236630; Abcam), and Calnexin (1:1,000; #2679; Cell Signaling Technology, Danvers, MA, USA) overnight at 4 °C. After washing with TBST, the membranes were treated with secondary antibodies (1:2,000; #7076; CST) for 60 min at room temperature. Protein bands were detected using ECL.

PKH26 staining assay was performed to defined exosomes presence. Briefly, 20 μL PKH26 dye (Sigma-Aldrich) was combined with 200 μL exosome suspension, incubating for 10 min. Then, process was then halted by adding 800 μL PBS to the mixture. After being collected by ultracentrifugation at 100,000 g for 70 min, the tagged exosomes were resuspended in 200 μL PBS. The presence of PKH26-labeled exosomes was examined through fluorescence microscope (Nikon, Japan).

## Oxygen-glucose deprivation (OGD) modeling and exosome treatment

Six-well plates with a seeding density of $4 \times 10^5$ BV-2 cells each were used for the 20 h of culturation. After that, the media was changed to glucose-free DMEM containing 2 mM sodium pyruvate and 1% FBS. The cells were then incubated for 4 h at 37 °C in a humidified environment that included 1% $O_2$, 5% $CO_2$, and 94% $N_2$. After OGD, the glucose-free medium switched out for normal DMEM, and cells were developed for 4 h under normoxic conditions. After modelling, cells were treated with 15 μg/mL exosome. Cells without any treatment were set as control group.

## Flow cytometry for apoptosis

To detect cell apoptosis using flow cytometry, cells were gathered by centrifugation and then homogenized in 1× Binding buffer at a density of $2 \times 10^5$ cells/mL. Annexin V-FITC (Sangon Biotech, Shanghai, China) was added and the mixture was incubated at 4 °C for 10 min. After washing and resuspending cells in 1× Binding buffer, propidium iodide was added. Cells were immediately examined through flow cytometry for double staining detection of apoptosis.

## Flow cytometry for cell phenotype

For flow cytometry analysis of the BV-2 cell phenotype, cells were harvested and resuspended in 100 μL of PBS. Anti-mouse CD40-FITC (#124607; Biolegend, San Diego, CA, USA), CD86-PE (#105007, Biolegend), and CD206-FITC (#141703, BioLegend) antibodies were incubated with cells at 4 °C for 30 min. Cells were resuspended in 500 μL of PBS after being washed twice with PBS. The samples were then evaluated using a flow cytometer (BD Biosciences, Franklin Lakes, NJ, USA).

## miRNA identification and analysis

RNA was extracted from BV-2 cells in OGD or OGD + exo groups. Using a NanoDrop spectrophotometer, concentration and purity of the RNA were assessed, and RNA integrity was confirmed through gel electrophoresis. Using Illumina HiSeq platform, six small RNA libraries were built and sequenced. By eliminating adaptor sequences, poor-quality reads, and reads smaller than 18 nt or bigger than 30 nt from the raw sequencing data, clean data was obtained. Bowtie was used to map the clean reads to the reference genome (Mus musculus), and miRDeep2 was utilized to determine the miRNA expression levels.

## Differentially expressed miRNA analysis

The edgeR package in R was used to carry out a differential expression analysis. Absolute $P < 0.05$ and $|log2FC| \geq 1$ were applied to define which miRNAs were differentially expressed (*Yang et al., 2022*). Volcano plots were generated to visualize the distribution of differentially expressed miRNAs. To investigate the link between various samples, a hierarchical clustering analysis was carried out using the R heatmap package.

## Functional annotation and enrichment analysis of miRNA targets

We used TargetScan, miRanda, and DIANA-microT algorithms to search for differentially expressed miRNA-related genes. Using the DAVID tool, functional annotation and enrichment analysis were performed on the anticipated targets. The predicted targets with score = 1 and validated target genes using qRT-PCR were subjected to enrichment analyses of Gene Ontology (GO, http://www.geneontology.org/), Kyoto Encyclopedia of Genes and Genomes (KEGG, http://www.genome.jp/kegg/), disease ontology (DO, http://www.disease-ontology.org), and Reactome (http://reactome.org/). Enrichment results were visualized using bubble plots. Adjust $p$ value $< 0.05$ was used as the cutoff criterion.

## Cell transfection

BV-2 cells were cultured in 6-well plates and transfected with miRNA mimics, inhibitors, or plasmids using Lipofectamine 3000. The miRNA mimics and inhibitors sequences

were supplied by GenePharma (Shanghai, China). Plasmids used in this study were psiCHECK-2-Traf6 (wild-type 3'UTR), psiCHECK-2-Traf6-mut (mutated 3'UTR), oe-Traf6 (overexpression of Traf6), and sh-Traf6-1 (shRNA against Traf6-1), sh-Traf6-2, and sh-Traf6-3.

### Dual luciferase reporter gene assay

BV-2 cells were collected after 48 h of co-transfection and washed with pre-chilled PBS. Careful washing was performed to prevent cell detachment and loss. $5\times$ PLB was diluted to $1\times$ PLB with deionized water. 100 μL $1\times$ PLB was added to each well of a white, opaque 96-well plate for cell lysis, and shaken on a shaker for 15 min at room temperature. After 20 μL cell lysate was added to each well, pre-mixed Stop&Glo Reagent (100 μL) was added, and luminescence data were measured after a 2 s delay. Luciferase activity detection was performed under light-avoiding conditions.

### RT-qPCR analysis

Total RNA was extracted using TRIzol reagent (Thermo Fisher Scientific, Waltham, MA, USA). The RNA concentration was determined using a NanoDrop spectrophotometer, and cDNA was synthesized using a reverse transcription kit (Thermo Fisher Scientific). The expression levels of mmu-miR-6238, mmu-miR-3102-3p, mmu-miR-6984-5p, mmu-miR-495-3p, mmu-miR-3095-3p, mmu-miR-6975-5p, mmu-miR-7013-5p, mmu-miR-7652-5p, and mmu-miR-146a-5p were detected by quantitative real-time PCR (qPCR) using the PowerUp SYBR Green Master Mix (Thermo Fisher Scientific) and specific primers (Table S1). The qPCR conditions were as follows: 95 °C for 10 min, followed by 40 cycles of 95 °C for 15 s and 60 °C for 20 s. U6 snRNA was used as an internal control. The expression levels were calculated using the $2^{-\Delta\Delta Ct}$ method.

### Statistical analysis

Statistical analysis was carried out using GraphPad Prism 8.0 software. Data are presented as mean ± standard deviation, and were analyzed using one-way ANOVA followed by Tukey's test or unpaired Student's $t$-test. $P$ value of less than 0.05 was considered statistically significant.

## RESULTS

### Exosomes were successfully isolated from BMSCs in mice

We isolated exosomes from BMSCs and characterized them using various methods. TEM images of the isolated exosomes revealed their typical cup-shaped morphology, indicating successful isolation (Fig. 1A). Particle size analysis demonstrated a relatively homogeneous population of exosomes, with an average particle size of 135.7 nm and concentration of $6.0 \times 10^{10}$ particles/mL (Fig. 1B). Western blotting detected presence of exosome markers TSG101 and CD9. Absence of Calnexin further indicated the purity of the exosome preparation (Fig. 1C). Additionally, a tracing assay using the PKH26 dye showed strong red fluorescence signals in the labeled exosomes, providing further evidence for the presence of exosomes in the isolated sample (Fig. 1D).

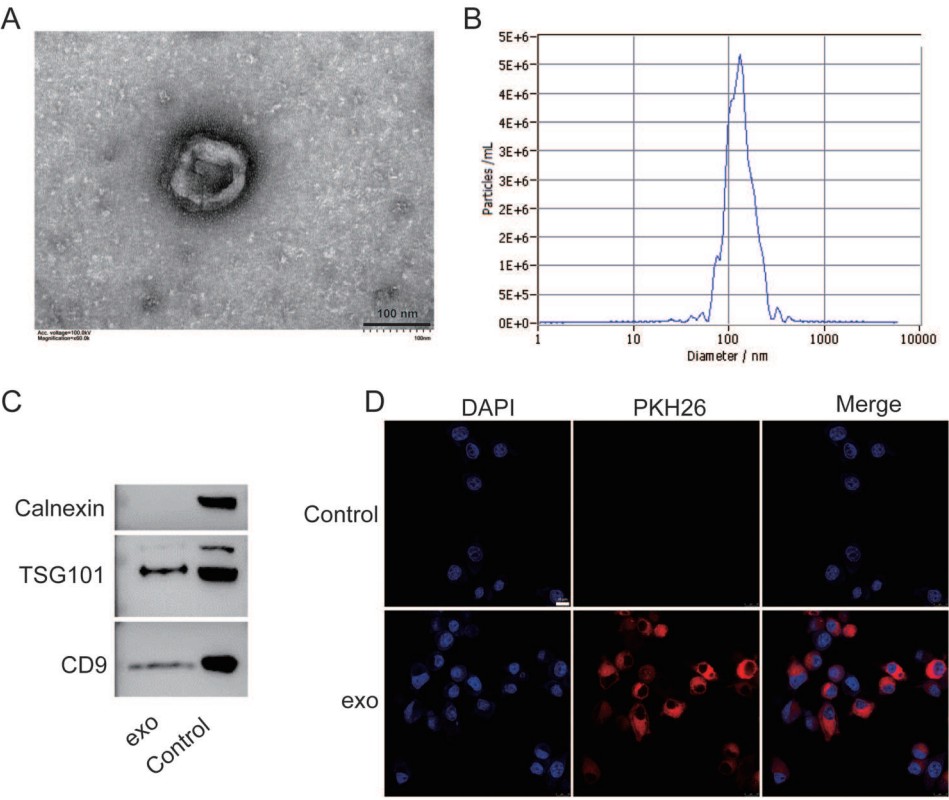

**Figure 1 Characterization of exosomes isolated from BMSCs.** (A) Transmission electron microscopy (TEM) image showing the cup-shaped morphology of isolated exosomes. Scale bar = 100 nm. (B) Particle size analysis of exosomes, indicating an average particle size of 135.7 nm and a concentration of $6.0 \times 10^{10}$ particles/mL. (C) Western blot analysis of exosome markers TSG101 and CD9, and the absence of endoplasmic reticulum marker Calnexin. (D) Tracing assay with PKH26 dye showing strong red fluorescence signals in labeled exosomes. exo: exosomes.

## BMSCs-derived exosomes modulated microglial conversion

We then investigated the function of BMSCs-derived exosomes on BV-2 microglial cell polarization. Results revealed that BV-2 cells displayed a typical activated phenotype with an amoeboid shape and retracted processes (Fig. 2A). In comparison with the control cells, apoptosis in OGD-exposed cells were elevated, and there was a significant increase of apoptosis in exosome-treated OGD cells compared with untreated cells ($P < 0.0001$, Fig. 2B). Furthermore, the expression of CD40 and CD86 (M1-like microglia markers) and CD206 (M2-like microglia marker) was assessed. Positive ratio of CD40 and CD86 was increased while that of CD206 was decreased in the OGD-exposed cells ($P < 0.05$, Fig. 2C). Exosome treatment led to significant upregulation of CD40 and CD86 while downregulation of CD 206 ($P < 0.05$, Fig. 2C).

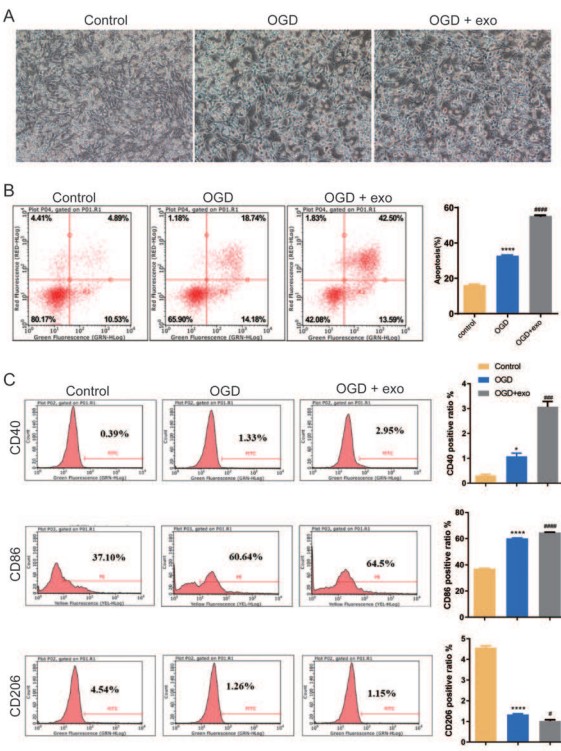

**Figure 2  Effects of BMSC-derived exosomes on microglial polarization.** (A) TEM image of BV-2 cells displaying an activated phenotype with an amoeboid shape and retracted processes. (B) Flow cytometry analysis of apoptosis in OGD-exposed BV-2 cells treated with exosomes. (C) Expression levels of CD40, CD86 (M1-like markers), and CD206 (M2-like marker) in exosome-treated and untreated cells. $*P < 0.05$, $****P < 0.0001$ *vs.* control groups; $\#P < 0.05$, $\#\#\#P < 0.001$, $\#\#\#\#P < 0.0001$ *vs.* OGD groups. exo: exosomes.

## Biology information analyses of miRNA in microglia
### miRNA identification

RNA was extracted from OGD-exposed microglial cells treated with or without exosomes and sequencing analyses was performed. Using the Illumina platform, the base quality scores were determined, and the base recognition accuracy rate was found to be greater than 90% (Table 1). The sequencing base quality in this study was primarily consistent with the Q30 criterion (Fig. 3A). We then employed miRDeep2 to identify miRNA expression in the samples. A total of six miRNAs were identified, including mmu-let-7c-2-3p, mmu-let-7a-1-3p, mmu-let-7a-5p (mmu-let-7a-1 and mmu-let-7a-2), mmu-let-7a-2-3p, and mmu-let-7b-3p. Subsequently, we performed hierarchical clustering analysis on the samples, and the dendrogram displayed the Euclidean distances between different samples, with a truncation height ranging from 10,000 to 80,000 (Fig. 3B). Principal component analysis (PCA) and heatmap visualization revealed substantial heterogeneity between the two different treatment groups (Fig. 3C and 3D).

**Table 1  Illumina sequencing quality value.**

| Phred score | Incorrect base identification ratio | Correct base identification ratio | Q.score |
|---|---|---|---|
| 10 | 1/10 | 90% | Q10 |
| 20 | 1/100 | 99% | Q20 |
| 30 | 1/1,000 | 99.9% | Q30 |
| 40 | 1/10,000 | 99.99% | Q40 |

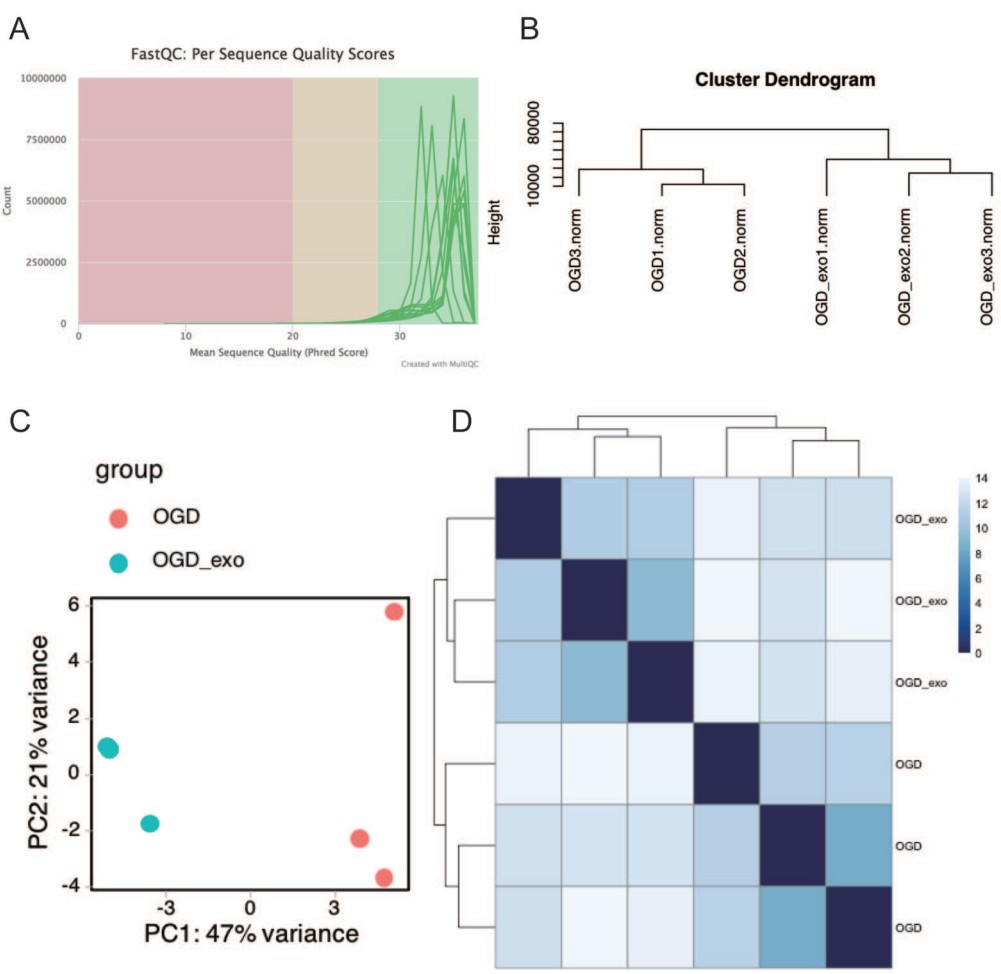

**Figure 3  Quality assessment and clustering analysis of miRNA-seq data.** (A) Base quality scores distribution for sequenced small RNA samples. (B) Dendrogram displaying Euclidean distances between samples based on hierarchical clustering analysis. (C) Principal component analysis (PCA) plot showing sample separation between treatment groups. (D) Heatmap visualization of miRNA expression profiles for the two treatment groups.

### Differentially expressed miRNA

We analyzed the length distribution of the identified expressed miRNAs, revealing a predominant length of approximately 21 bp (Fig. 4A). Differential expression analysis on these miRNAs was performed using a selection criterion of $P < 0.05$ and $|log2FC| \geq 1$. The volcano plot showed that exosome treatment led to significant changes in miRNA expression in OGD-exposed microglial cells, with 10 upregulated miRNAs and 33 downregulated miRNAs (Fig. 4B). The differentially expressed miRNAs were visualized using a heatmap and a bar chart; The red color is upregulated miRNAs, while the blue color represents downregulated miRNAs in the heatmap (Fig. 4C). The bar chart displayed mature and precursor names of these differentially expressed miRNAs (Fig. 4D).

### Enrichment analyses of miRNA

We searched online for targets of differentially expressed miRNAs. The details of the targets were shown in Tables S2 and S3. The predicted target genes for mmu-miR-429-3p included Fez2 and Fignl2; for mmu-miR-142a-3p, Utrn and Rock2; for mmu-miR-340-5p, Kctd14; and for mmu-miR-429-3p, Rps6kb1. For experimentally validated differentially expressed miRNAs, the results revealed that mmu-miR-155-5p targeted genes such as Sfpi1, Myb, Rheb, Bat5, Jarid2, and Trp53inp1. Subsequently, functional enrichment analysis was conducted on both the predicted target genes with a score value of 1 and experimentally validated target genes, and the top 20 terms were visualized using bubble plots.

For the predicted target genes, GO enrichment analysis indicated their significant association with processes such as pre-mRNA intronic binding, vinculin binding, and DNA-binding transcription repressor activity (Fig. S1A). KEGG pathway analysis revealed their involvement in Yersinia infection, Leishmaniasis, and Pertussis (Fig. S1B), while Reactome pathways primarily included p75NTR recruitment of signaling complexes, NF-kB activation and survival signaling, and IRAK1 recruitment of the IKK complex (Fig. S1C).

For the experimentally validated target genes, GO terms were enriched in processes such as muscle cell proliferation, regulation of smooth muscle cell proliferation, and smooth muscle cell proliferation (Fig. S2A). KEGG pathway analysis indicated their involvement in osteoclast differentiation, hepatitis B, lipid metabolism, and atherosclerosis (Fig. S2B). DO terms were associated with T-cell leukemia, lymphoid leukemia, and bacterial infectious diseases (Fig. S2C). Reactome pathways were mainly related to cytokine signaling in the immune system, MyD88-independent TLR4 cascade, and TRIF(TICAM1)-mediated TLR4 signaling (Fig. S2D).

## BMSCs-derived exosomes mmu-miR-146a-5p modulated microglial conversion

Based on the sequencing results, we selected mmu-miR-6238, mmu-miR-3102-3p, mmu-miR-6984-5p, mmu-miR-495-3p, mmu-miR-3095-3p, mmu-miR-6975-5p, mmu-miR-7013-5p, mmu-miR-7652-5p, and mmu-miR-146a-5p for validation. The results demonstrated that in OGD-exposed microglial cells, only mmu-miR-6984-5p expression was upregulated after exosome treatment, but the change was not statistically

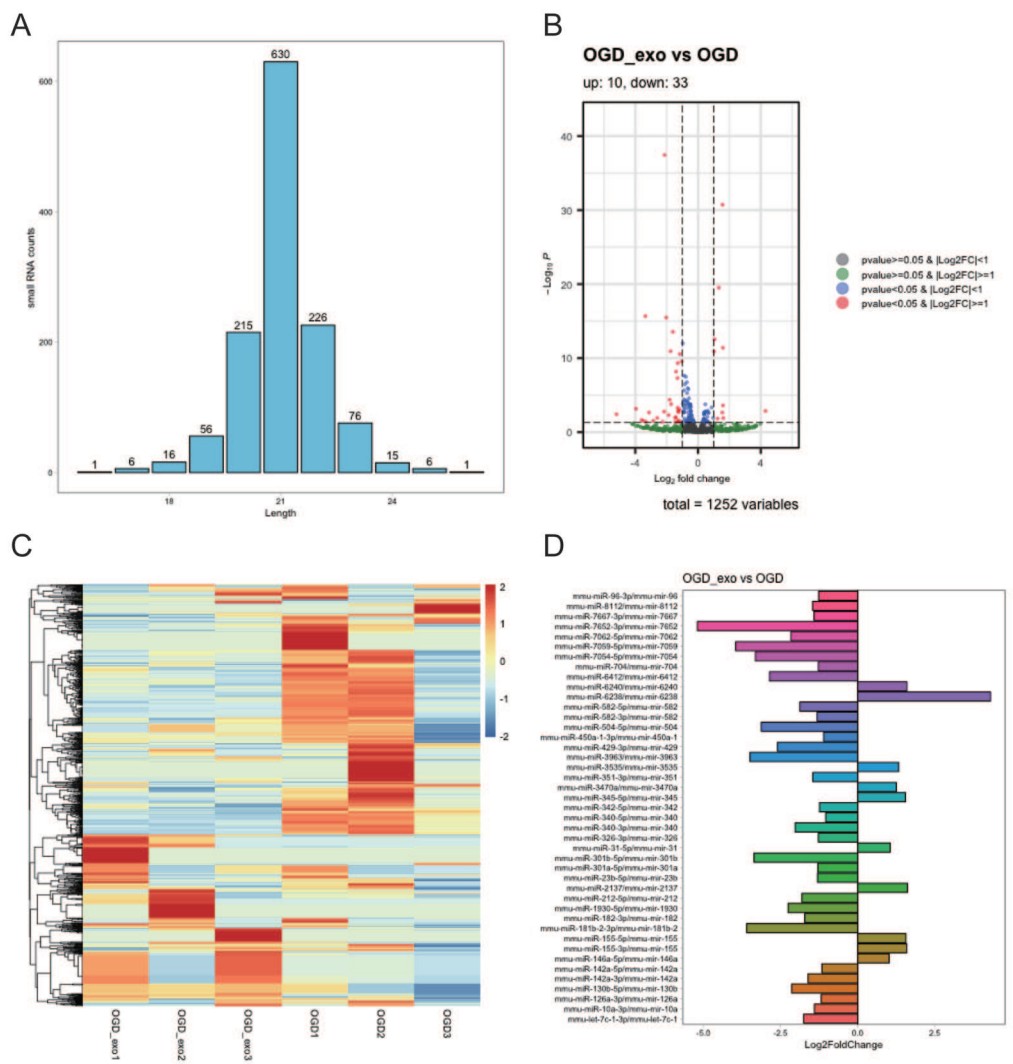

**Figure 4  Identification of differentially expressed miRNAs.** (A) Length distribution of identified expressed miRNAs. (B) Volcano plot showing upregulated (red) and downregulated (blue) miRNAs in exosome-treated OGD-exposed microglial cells. Selection criterion: $P < 0.05$ and $|\log 2FC| \geq 1$. (C) Heatmap visualization of differentially expressed miRNAs. Red: upregulated; blue: downregulated. (D) Bar chart displaying the mature and precursor names of differentially expressed miRNAs.

significant (Fig. 5). Expression of mmu-miR-3102-3p, mmu-miR-3095-3p, mmu-miR-6975-5p, mmu-miR-7013-5p, mmu-miR-7652-5p, and mmu-miR-146a-5p was significantly decreased ($P < 0.05$), with mmu-miR-146a-5p showing the most prominent downregulation ($P < 0.0001$, Fig. 5). Therefore, we chose mmu-miR-146a-5p for further analysis.

To explore the role of mmu-miR-146a-5p on microglial conversion, OGD-exposed BV-2 cells were transfected with miR-146a-5p mimics or inhibitors. The cell morphology

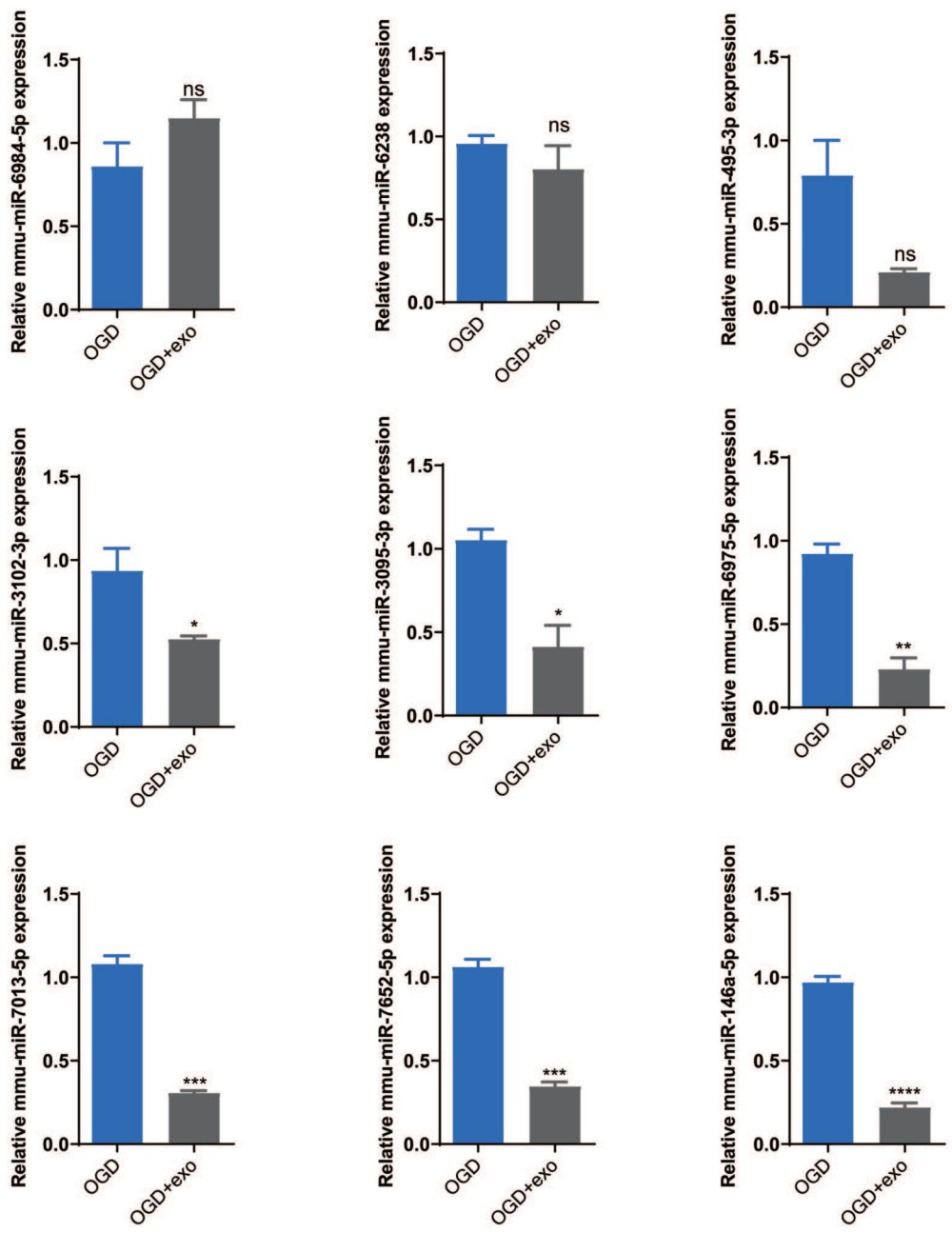

**Figure 5  Validation of selected miRNAs.** Expression levels of selected miRNAs (mmu-miR-6238, mmu-miR-3102-3p, mmu-miR-6984-5p, mmu-miR-495-3p, mmu-miR-3095-3p, mmu-miR-6975-5p, mmu-miR-7013-5p, mmu-miR-7652-5p, and mmu-miR-146a-5p) in OGD-exposed microglial cells treated with or without exosomes. *$P < 0.05$, **$P < 0.01$, ***$P < 0.001$, ****$P < 0.0001$ *vs.* OGD groups. ns: no significance; exo: exosomes.

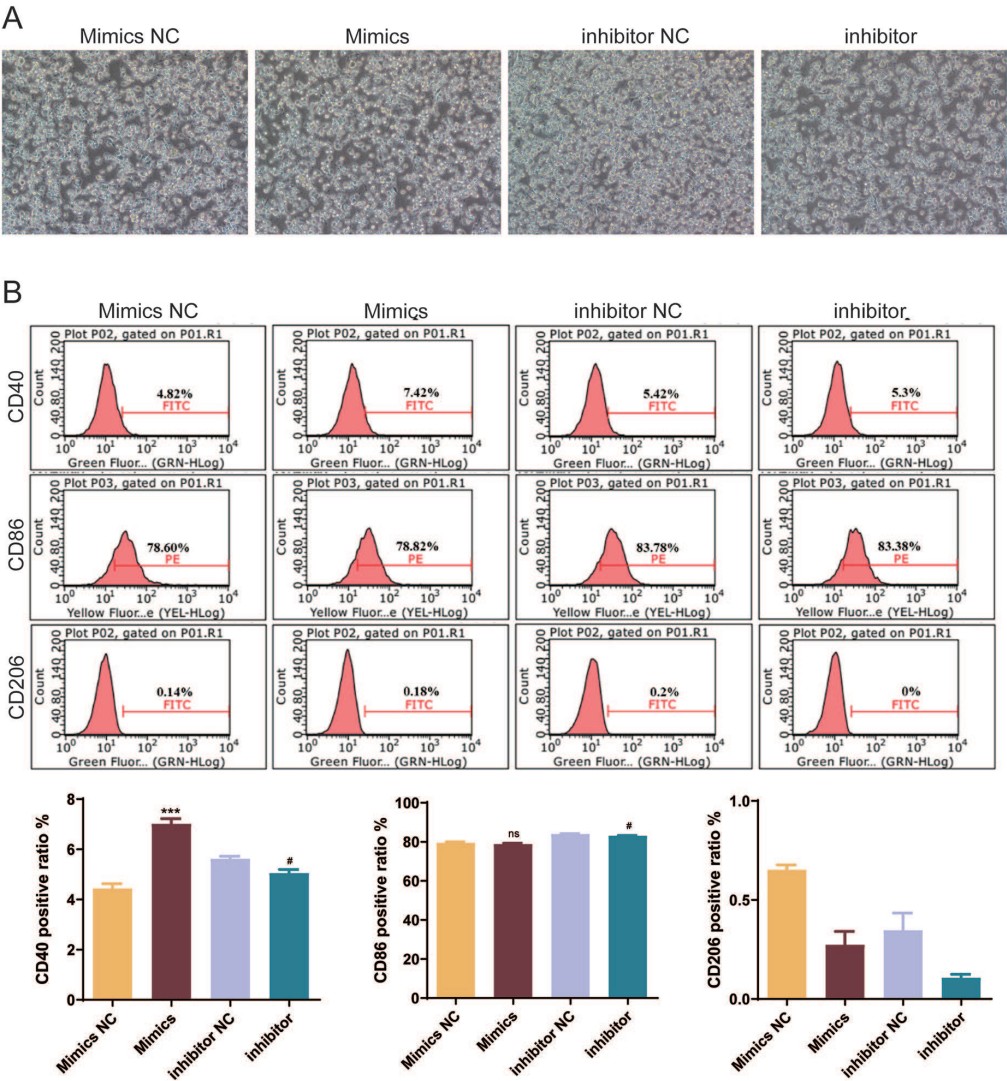

**Figure 6  Effects of mmu-miR-146a-5p on microglial conversion.** (A) Cell morphology diagram of BV-2 cells transfected with mmu-miR-146a-5p mimics or inhibitors. (B) Expression of CD40, CD86, and CD206 in BV-2 cells transfected with mmu-miR-146a-5p mimics or inhibitors. ***$P < 0.001$ *vs.* Mimics NC groups; #$P < 0.05$ *vs.* inhibitor NC groups. ns: no significance.

diagram shows that the treated cells are growing well (Fig. 6A). After upregulated miR-146a-5p, CD40 positive ratio was increased compared with mimics NC groups ($P < 0.001$, Fig. 6B). Oppositely, miR-146a-5p inhibitors reduced the number of CD40 and CD86 cells compared to inhibitor NC groups ($P < 0.05$, Fig. 6B). The number of CD206-positive cells showed no significant changes after transfection (Fig. 6B).

### BMSCs-derived exosomes mmu-miR-146a-5p modulated microglial conversion by inhibiting Traf6

Subsequently, we validated the downstream targets of mmu-miR-146a-5p. mmu-miR-146a-5p mimics was transfected into OGD BV-2 cells. Compared to miR NC, mmu-miR-146a-5p expression significantly increased after transfection ($P < 0.0001$), indicating successful transfection of mmu-miR-146a-5p into cells (Fig. 7A). In contrast, the Traf6 mut group did not affect mmu-miR-146a-5p expression compared to psiCHEK-2-Traf6 (Fig. 7A). Dual-luciferase reporter assay revealed that mmu-miR-146a-5p reduced luciferase activity ($P < 0.0001$), suggesting that mmu-miR-146a-5p could target Traf6 3′UTR (Fig. 7B). We then verified the regulatory relationship between mmu-miR-146a-5p and Traf6 in BV-2. Results showed that transfection of mmu-miR-146a-5p mimics decreased Traf6 expression in BV-2, while mmu-miR-146a-5p inhibitors upregulated Traf6 expression (Fig. 7C), indicating a negative regulatory relationship between mmu-miR-146a-5p and Traf6.

We further explored role of the mmu-miR-146a-5p/Traf6 axis on microglial cell polarization. qPCR results demonstrated that sh-Traf6-1 and sh-Traf6-2 significantly downregulated Traf6 expression ($P < 0.01$, Fig. 7D) and oe-Traf6-1 significantly elevated the expression of Traf6 ($P < 0.01$, Fig. 7E). Therefore, BV-2 cells were then co-transfected with mmu-miR-146a-5p mimics and oe-Traf6-1 or mmu-miR-146a-5p inhibitor and sh-Traf6-1/2, and all cell groups showed good growth (Fig. 7F). No significant changes were found in CD40 positive ratio (Fig. 7G). miR-146a-5p mimics and oe-Traf6 increased the number of CD86-positive cells while decreasing CD206-positive cells ($P < 0.0001$, Fig. 7G). In contrast, mmu-miR-146a-5p inhibitor and sh-Traf6-1/2 reduced the number of CD86-positive cells while increasing CD206-positive cells ($P < 0.0001$, Fig. 7G).

## DISCUSSION

Exosomes are vital mediators for intercellular communication, and BMSCs-derived exosomes have been demonstrated to have therapeutic effects in various diseases (*Xiao et al., 2023*). Herein, we successfully isolated exosomes from BMSCs and determined their size and purity. The results revealed that BMSC-derived exosomes effectively modulated microglial conversion, with a significant increase in apoptosis and upregulation of M1-like microglia markers CD40 and CD86, and downregulation of M2-like microglia marker CD206. miRNA sequencing analysis identified several differentially expressed miRNAs, among which miR-146a-5p was significantly downregulated after exosome treatment. Further experiments revealed that miR-146a-5p modulated microglial polarization by targeting Traf6.

Depending on the activation state, microglia can display different phenotypes. The classic activated, M1 phenotype is characterized by expression of markers such as CD40, CD86, and iNOS, and the release of pro-inflammatory cytokines including TNF-α, IL-1β, and IL-6 (*Ban et al., 2023*). In contrast, the alternatively activated, M2 phenotype is characterized by expression of markers such as CD206, Arg1, and YM1, and the release of anti-inflammatory cytokines such as IL-4, IL-10, and TGF-β (*Chen et al., 2023*). Balance
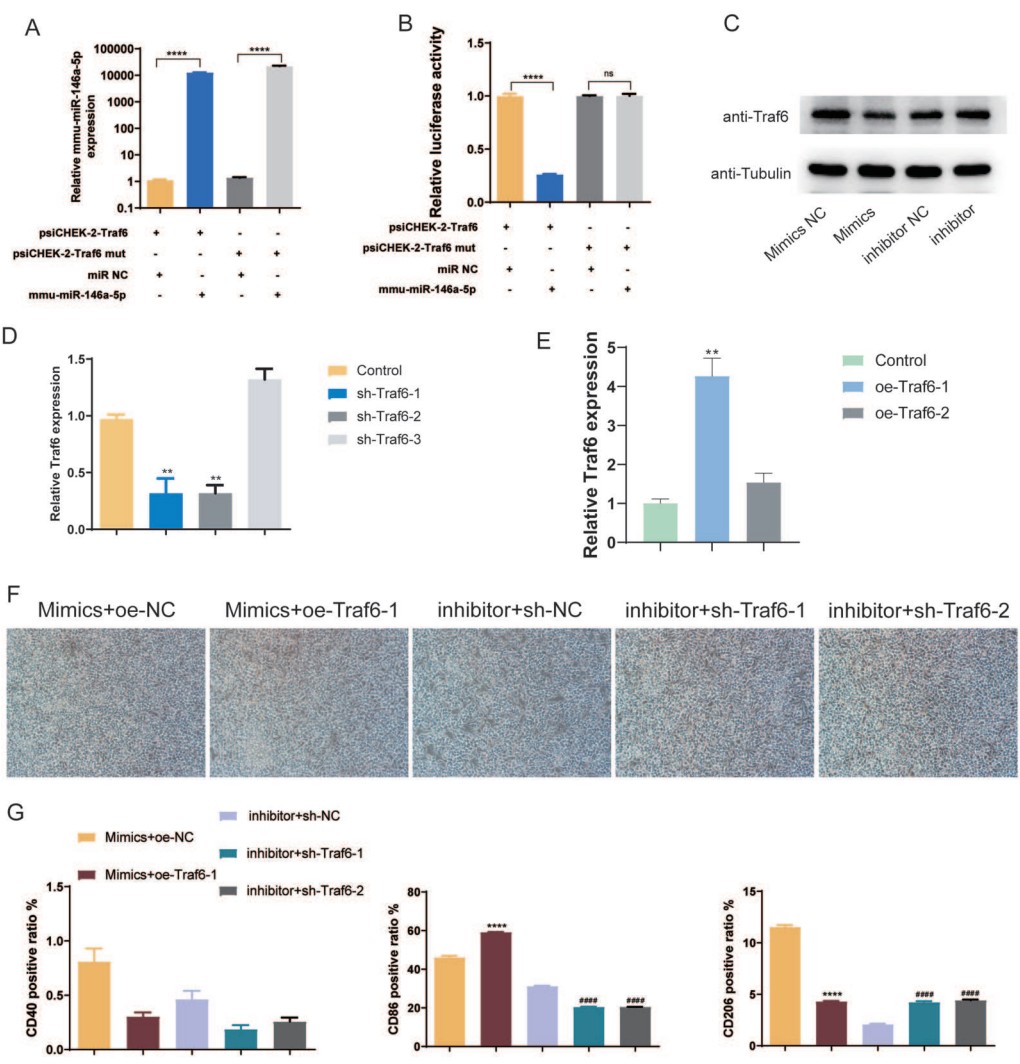

**Figure 7** **Role of mmu-miR-146a-5p/Traf6 axis in microglial conversion.** (A) Expression of mmu-miR-146a-5p and Traf6 in transfected OGD BV-2 cells. ****$P < 0.0001$. (B) Dual-luciferase reporter assay showing the targeting of Traf6 3′UTR by mmu-miR-146a-5p. ****$P < 0.0001$; ns: no significance. (C) Western blot analysis of Traf6 expression in BV-2 cells transfected with mmu-miR-146a-5p mimics or inhibitors. (D–E) RT-qPCR analysis of Traf6 expression. **$P < 0.01$ *vs.* control groups. (F) Cell growth after transfection with mmu-miR-146a-5p mimics and oe-Traf6 or mmu-miR-146a-5p inhibitor and shTraf6-1/2. (G) Expression of CD40, CD86, and CD206 in BV-2 cells transfected with mmu-miR-146a-5p mimics and oe-Traf6 or mmu-miR-146a-5p inhibitor and shTraf6-1/2. ****$P < 0.0001$ *vs.* Mimics + oe-NC groups; ####$P < 0.0001$ *vs.* inhibitor + sh-NC groups.

between M1 and M2 microglia plays a critical role in various CNS disorders (*Li et al., 2023a*). Our data are consistent with previous studies that have shown ability of exosomes to modulate microglial activation and polarization. For example, *Liu et al. (2022)* reported that human umbilical cord mesenchymal stem cells-derived exosomes were able to suppress the activation of M1-like microglia and promote the polarization of M2-like microglia *in vitro* and *in vivo*. Our findings showed that BMSC-derived exosomes decreased CD40

and CD86 while increasing CD206, indicating polarization of microglia towards an M2 phenotype.

Exosomes contain various bioactive molecules, including miRNAs, which can be transferred between cells and modulate cellular functions. We identified differential expression of exosomal miRNAs isolated from OGD-exposed BV-2 cells by sequencing analysis, in which miR-146a-5p was significantly down-regulated. miR-146a-5p is a member of the miR-146 family, which is a well-studied group of miRNAs involved in regulation of immune responses and inflammation (*Lu et al., 2018*). It is mainly expressed in various immune cells, including monocytes, macrophages, and dendritic cells, and has been shown to play a crucial role in modulating the polarization of macrophages (*Deng et al., 2022*; *Lai et al., 2022*). Several studies have revealed potential therapeutic applications of miR-146a-5p in various diseases, including inflammation-related diseases such as arthritis, atherosclerosis, and sepsis (*Huang et al., 2021*; *Li, Liu & Huang, 2023*; *Yu et al., 2022*; *Zheng et al., 2022*). miR-146a-5p has been found to promote the M2-like phenotype of macrophages, which is characterized by an anti-inflammatory and tissue-repairing function (*Huang, Lin & Lin, 2023*). Conversely, it inhibits the M1-like phenotype of macrophages, which is associated with pro-inflammatory responses and tissue destruction. *Duan et al. (2020)* demonstrated that miR-146a-5p inhibited microglial M1 polarization in intracerebral hemorrhage rats. Our results showed that BMSC-derived exosomes containing miR-146a-5p could modulate microglial polarization toward the M2-like phenotype, as indicated by the upregulation of CD206 and downregulation of CD40 and CD86.

Furthermore, our study revealed that miR-146a-5p directly targeted the Traf6 gene, which is a signaling adaptor protein involved in several cellular processes, including inflammation, immune response, and cell survival (*Fang & Hong, 2021*). In the context of microglial polarization, Traf6 plays a critical role in M1-like microglial activation and polarization by regulating NF-κB signaling and downstream pro-inflammatory cytokine production (*Wen et al., 2022*). On the other hand, inhibiting Traf6 activity can promote M2-like microglial polarization and reduce inflammation (*Cheng et al., 2023*). Studies have reported that miR-146a targets Traf6 and negatively regulates its expression in microglial (*Liu et al., 2020*). Overexpression of miR-146a-5p could promote M2-like microglial polarization and reduce inflammation by inhibiting the Traf6/NF-κB signaling, while inhibiting miR-146a-5p has the opposite effect, promoting M1-like microglial polarization and inflammation (*Ge et al., 2019*; *Zhang et al., 2021*). Our findings were consistent with these previous studies, and in OGD-exposed BV-2 cells, miR-146a-5p negatively regulates Traf6 and inhibits cell polarization towards the M1 phenotype. These findings suggest that miR-146a-5p/Traf6 may have therapeutic potential in the treatment of neuroinflammatory diseases by modulating microglial polarization.

Despite this study provides important insights into the role of BMSC-derived exosomes and miR-146a-5p in modulating microglial polarization, there are several limitations that should be considered. Firstly, the study does not directly assess the functional consequences of these changes, which need further exploration. Secondly, our study focused on miR-146a-5p due to its significant downregulation after exosome treatment, but other differentially expressed miRNAs identified in the study may also play important roles in modulating

microglial polarization. Thirdly, the direct relationship between exosomes and miR-146a-5p has not been explored in the present study. Further studies investigating the function of these miRNAs could provide a more comprehensive understanding of the molecular mechanisms involved in microglial polarization.

## CONCLUSION

In conclusion, our study demonstrated that BMSC-derived exosomes effectively modulated microglial conversion, with a significant increase in apoptosis and upregulation of M1-like microglia markers CD40 and CD86, and downregulation of M2-like microglia marker CD206. We identified several differentially expressed miRNA exosomes, among which miR-146a-5p was downregulated the most significantly. Further experiments demonstrated that miR-146a-5p modulated microglial polarization by targeting Traf6. Our findings provide novel insights into the role of exosomal miRNAs in modulating microglial polarization and highlight the potential therapeutic applications of BMSC-derived exosomes in treating neuroinflammatory diseases.

### Funding
This work was supported by the Natural Science Foundation of Fujian Province (No.2020J011227). The funders had no role in study design, data collection and analysis, decision to publish, or preparation of the manuscript.

### Grant Disclosures
The following grant information was disclosed by the authors:
The Natural Science Foundation of Fujian Province: No.2020J011227.

### Competing Interests
The authors declare there are no competing interests.

### Author Contributions
- Xianwei Huang conceived and designed the experiments, performed the experiments, analyzed the data, prepared figures and/or tables, and approved the final draft.
- Xiong Liu conceived and designed the experiments, analyzed the data, prepared figures and/or tables, and approved the final draft.
- Jiaqi Zeng performed the experiments, analyzed the data, authored or reviewed drafts of the article, and approved the final draft.
- Penghui Du performed the experiments, analyzed the data, authored or reviewed drafts of the article, and approved the final draft.
- Xiaodong Huang performed the experiments, analyzed the data, authored or reviewed drafts of the article, and approved the final draft.
- Jiyan Lin conceived and designed the experiments, authored or reviewed drafts of the article, and approved the final draft.

## Data Availability

The sequence data is available at NCBI SRA: SAMN36999226, SAMN36999227, SAMN36999228, SAMN36999229, SAMN36999230, SAMN36999231.

The data is available in the Supplemental Files and at Zenodo: Lin Jiyan. (2023). Bone marrow mesenchymal stem cell-derived exosomal microRNA regulates microglial polarization [Data set]. Zenodo. https://doi.org/10.5281/zenodo.8213321.

## Supplemental Information

Supplemental information for this article can be found online at http://dx.doi.org/10.7717/peerj.16359#supplemental-information.

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
