# Peer review of "Bone marrow mesenchymal stem cell-derived exosomal microRNA regulates microglial polarization"

_PeerJ, doi:10.7717/peerj.16359_

## Round 0.1 · original submission · Minor Revisions

Please carefully read the comments and suggestions from the reviewer and provide your point-by-point responses.

Reviewer 1 ·

Basic reporting

1.In this article, presentations about the comparison between groups is not indicated precisely in plenty of Figures. Please make corrections if necessary.
2.Please check the fonts style for writing. For example, figure legend sections should adopt ‘Times New Roman’ font style.
3.In line 243, line 303, and line 335, ‘DO’ , ‘dataare’, and ‘microlial’ maybe spelled incorrectly. Please make corrections if necessary.
4.There exists some grammar problems for Engish expression, such as line 116, line 139, line 302, and line 355. Please make corrections if necessary.
5.The figure style seems a little inconsistent. For example, the group legendwas expressed as “Mimics” (capital) in Figure 5, while this word was expressed as “mimics” (lower case) in Figure 6, and the inconsistency was found in “Control” and “control”. Additionally, “sh-Traf6” was used to denote Traf6 knockdown, however, which was expressed as “shTraf6” in figures. Please keep the expression style consistent.
6.Please check the figure legend section. For example, “**P< 0.01” did not comply to Figure 2.
7.Please provide related references to support the statement “Exosomes are vital mediators for intercellular communication, and BMSCs-derived exosomes have been demonstrated to have therapeutic effects in various diseases” in the discussion section.

Experimental design

1.The ‘Material and Methods’ should be illustrated thoroughly. For example, there are some deficiencies in declaring how the BV-2 cell line was obtained. And primary cells and cell lines were not mentioned and distinguished. Please add more details to resolve this problem. Meanwhile, the URLs of tools used for GO and KEGG enrichment analysis should be provided. Please add necessary details.
2.Why was the criterion for differentially expressed miRNAs mentioned in the materials &methods section “|log2FC| > 1 and a false discovery rate (FDR) 0.05”( in line 143) inconsistent with that in the result and figure 4 legend sections (P< 0.05 and |log2FC| ≥ 1)? Please explain it and provide references to support the actual criterion applied in this part.
3.How to prove that stale Traf6-overexpressed cell lines were successfully established in this study (Figure 7)?

Validity of the findings

This study combined bioinformatic methods with RNA sequencing analysis, illustrating mechanisms of BMSC-derived exosomes modulate microglial polarization. Subsequently, the result provides a potential therapeutic target for the treatment of neurological diseases involving microglial activation in clinics.

Reviewer 2 ·

Basic reporting

This study is very interesting and meaningful, which found that BMSC-derived exosome modulated microglial polarization which providing a potential therapeutic target for the treatment of neurological diseases involving microglial activation.
The research design is standardized, the data statistical methods are appropriate, and the English expression is clear. The research findings warrant further research.

Experimental design

How is exosome treatment implemented in Line 114? When will exosomes be added? How much did you add? How is the control group set up?
Exosomes have complex components. How can we confirm a direct relationship between exosomes and mir-146a-5p?,
Why not establish non OGD control or non OGD plus exosomes control in group experiments?

Validity of the findings

Is there an internal reference in Figure 1 for C and WB to prove that the sample size is consistent? Is the scale bar in the D image consistent across all images?
Figure 2 A shows whether the magnification of Control, OGD, and OGD plus exosomes are consistent, and whether there are significant differences in cell morphology
Why is astrocytes not identified in Figure 6? How pure is it? Have OGD experiments been conducted after transfection with mir-146a-5p? Previous studies have confirmed that OGD itself promotes the transformation of microglia into M1 phenotype, and the effect of using MIMIC or inhibitor alone is not very significant, although the data differences are statistically significant.
The conclusion description in the abstract is not accurate and needs to be modified. It should be BMSC derived exosomal via miR-146a-5p modulates microglial polarization by targeting Traf6, providing a potential thermal target for the treatment of neurological diseases involving microglial activation

Additional comments

none

---

## Round 0.2 · accepted · Accept

The authors have provided their point-by-point responses to the comments and suggestions from the reviewers and made corrections in their manuscript accordingly. The paper may be accepted for publication.

Reviewer 1 ·

Basic reporting

no comment

Experimental design

no comment

Validity of the findings

no comment

Reviewer 2 ·

Basic reporting

After revision, the quality of the manuscript has been significantly improved. The authors have carefully responded to the questions I am concerned about one by one.

Experimental design

The research design is standardized, the data statistical methods are appropriate, and the English expression is clear.

Validity of the findings

This study is very interesting and meaningful, which found that BMSC-derived exosome modulated microglial polarization which providing a potential therapeutic target for the treatment of neurological diseases involving microglial activation. The research findings warrant further research.

Additional comments

none